# High-Throughput Synchronous Deep RL

**Iou-Jen Liu,  Raymond A. Yeh, Alexander G. Schwing**
University of Illinois at Urbana-Champaign
{iliu3, yeh17, aschwing}@illinois.edu

## Abstract

Deep reinforcement learning (RL) is computationally demanding and requires
processing of many data points. Synchronous methods enjoy training stability
while having lower data throughput. In contrast, asynchronous methods achieve
high throughput but suffer from stability issues and lower sample efficiency due
to 'stale policies.' To combine the advantages of both methods we propose High-
Throughput Synchronous Deep Reinforcement Learning (HTS-RL). In HTS-RL,
we perform learning and rollouts concurrently, devise a system design which
avoids 'stale policies' and ensure that actors interact with environment replicas
in an asynchronous manner while maintaining *full determinism*. We evaluate
our approach on Atari games and the Google Research Football environment.
Compared to synchronous baselines, HTS-RL is $2 - 6\times$ faster. Compared to
state-of-the-art asynchronous methods, HTS-RL has competitive throughput and
consistently achieves higher average episode rewards.

## 1  Introduction

Deep reinforcement learning (RL) has been impressively successful on a wide variety of tasks,
including playing of video games [1, 6, 12, 13, 20, 21, 23–25, 32] and robotic control [10, 18, 22].
However, a long training time is a key challenge hindering deep RL to scale to even more complex
tasks. To counter the often excessive training time, RL frameworks aim for two properties: (1) A
high throughput which ensures that the framework collects data at very high rates. (2) A high sample
efficiency which ensures that the framework learns the desired policy with fewer data. To achieve
both, synchronous and asynchronous parallel actor-learners have been developed which accelerate
RL training [1, 6–8, 19, 25, 29, 34].

State-of-the-art synchronous methods, such as synchronous advantage actor critic (A2C) [6, 25]
and related algorithms [26, 27, 35] are popular because of their *data efficiency*, *training stability*,
*full determinism*, and *reproducibility*. However, synchronous methods suffer from idle time as
all actors need to finish experience collection before trainable parameters are updated. This is
particularly problematic when the time for an environment step varies significantly. As a result,
existing synchronous methods don't scale to environments where the step time varies significantly
due to computationally intensive 3D-rendering and (physics) simulation.

Instead, asynchronous methods, *e.g.*, asynchronous advantage actor-critic on a GPU (GA3C) [1] and
importance weighted actor-learner architectures (IMPALA) [7], achieve high throughput. However,
these methods suffer from a stale-policy issue [1, 7] as learning and data collecting are performed
asynchronously. The policy which is used to collect data (behavior policy) is several updates behind
the latest policy (target policy) used for parameter updates. This lag leads to noisy gradients and thus
lower data efficiency. Consequently, asynchronous methods trade training stability for throughput.

We show that this trade-off is not necessary and propose High-Throughput Synchronous RL (HTS-
RL), a technique, which achieves both high throughput and high training stability. HTS-RL can
be applied to many off-the-shelf deep RL algorithms, such as A2C, proximal policy optimization

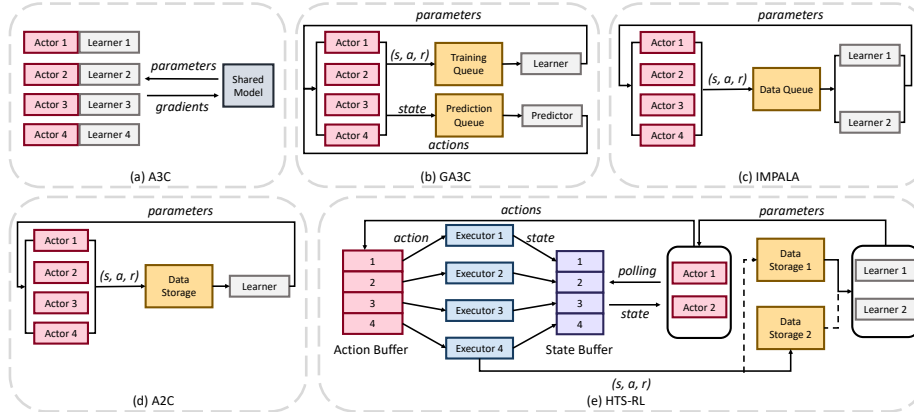

Figure 1: Structure and flow of (a) A3C, (b) GA3C, (c) IMPALA, (d) A2C and (e) our HTS-RL.

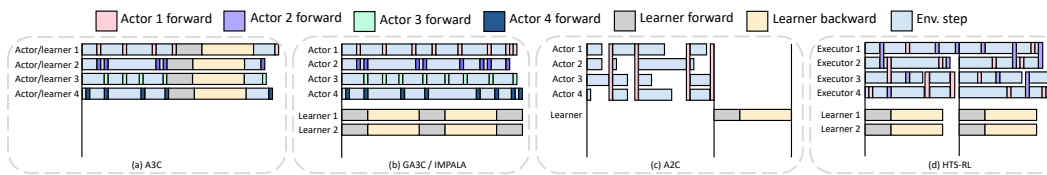

Figure 2: Processing timeline of (a) A3C , (b) GA3C/IMPALA, (c) A2C and (d) our HTS-RL.

(PPO) [27], and actor-critic using Kronecker-factored trust region (ACKTR) [35]. HTS-RL is particularly suitable for environments with large step time variance: It performs concurrent learning and data collection, which significantly increases throughput. Additionally, actors interact with environment replicas in an asynchronous manner. The asynchronous interaction reduces the idle time. Lastly, by performing batched synchronization, HTS-RL ensures that the lag between target and behavior policy equals one. Thanks to this constant latency, HTS-RL uses a 'one step delayed-gradient' update which has the same convergence rate as a non-delayed version. As a result, HTS-RL maintains the advantages of synchronous RL, *i.e.*, *data efficiency*, *training stability*, *full determinism*, and *reproducibility*, while achieving speedups, especially in environments where the step time varies.

We show that HTS-RL permits to speedup A2C and PPO on Atari environments [2, 3] and the Google Research Football environment (GFootball) [15]. Following the evaluation protocol of Henderson et al. [11] and Colas et al. [4], we compare to the state-of-the-art, asynchronous method, IMPALA [7]: our approach has $3.7\times$ higher average episode reward on Atari games, and $43\%$ higher average game scores in the GFootball environment. When compared with synchronous A2C and PPO, our approach achieves a $2-6\times$ speedup. Code is available at `https://github.com/IouJenLiu/HTS-RL`.

## 2  Related Work

In the following we briefly review work on asynchronous and synchronous reinforcement learning. **Asynchronous reinforcement learning:** Asynchronous advantage actor-critic (A3C) [25] is an asynchronous multi-process variant of the advantage actor-critic algorithm [30]. A3C runs on a single machine and does not employ GPUs. As illustrated in Fig. 1(a), in A3C, each process is an actor-learner pair which updates the trainable parameters asynchronously. Specifically, in each process the actor collects data by interacting with the environment for a number of steps. The learner uses the collected data to compute the gradient. Then the gradient is applied to a shared model which is accessible by all processes. One major advantage of A3C: throughput increases almost linearly with the number of processes as no synchronization is used. See Fig. 2(a) for an illustration of the timing.

Building upon A3C, Babaeizadeh et al. [1] further introduce the GPU/CPU hybrid version GA3C. However, GA3C suffers from the stale-policy issue: the behavior policy is several updates behind the target policy. This discrepancy occurs because GA3C decouples actors and learners as illustrated in Fig. 1(b) and Fig. 2(b), *i.e.*, actors and learners use different processes and perform update and data

collection asynchronously. Because of this discrepancy, on-policy algorithms like actor-critic become off-policy and the target policy may assign a very small probability to an action from the rollout data generated by the behavior policy. As a result, when updating the parameters, the log-probability may tend to infinity, which makes training unstable [1, 7]. To mitigate this, GA3C uses $\epsilon$-correction.

Importance weighted actor-learner architectures (IMPALA) [7] (Fig. 1(c) and Fig. 2(b)) adopt a system similar to GA3C. Hence, IMPALA also suffers from a stale-policy issue, harming its performance. In IMPALA, the behavior policy can be tens or hundreds of updates behind the target policy [7]. To correct this stale-policy issue, instead of $\epsilon$-correction, IMPALA introduces the importance sampling-based off-policy correction method 'V-trace.'

In contrast to those baselines, we develop High-Throughput Synchronous RL (HTS-RL), which is a fully deterministic synchronous framework. It enjoys the advantages of synchronous methods, *i.e.*, the impact of a stale-policy issue is minimal. More specifically, the behavior policy is guaranteed to be only one update behind the target policy. Thanks to the small delay, we avoid a stale-policy. As a result, HTS-RL doesn't require any correction method while being stable and data efficient.

**Synchronous reinforcement learning:** Synchronous advantage actor critic (A2C) [6] operates in steps as illustrated in Fig. 1(d) and synchronizes all actors at each environment step. As illustrated in Fig. 2(c), A2C waits for each actor to finish all its environment steps before performing the next action. A2C has advantages over its asynchronous counterpart: (1) It can more effectively use GPUs, particularly when the batch size is large; (2) It is fully deterministic, which guarantees reproducibility and facilitates development and debugging; (3) It does not suffer from stale policies; and (4) It has a better sample complexity [35].

Because of the aforementioned advantages, synchronous methods are popular. In addition to A2C, OpenAI Baselines [6] implement a series of synchronous versions of RL algorithms, including actor-critic using Kronecker-factored trust region (ACKTR) [35], trust region policy optimization (TRPO) [26], proximal policy optimization algorithms (PPO) [27], and sample efficient actor-critic with experience replay (ACER) [33]. However, one major drawback is a lower throughput compared to their asynchronous counterparts. This issue is exacerbated in environments where the time to perform a step varies.

To address this concern, we develop High-Throughput Synchronous RL (HTS-RL). HTS-RL performs concurrent learning and data collection, which significantly enhances throughput. Moreover, HTS-RL ensures that actors interact with environment replicas asynchronously while maintaining full determinism. To further reduce actor idle time, HTS-RL performs batched synchronization. Compared to A2C, which synchronizes every step (see Fig. 2(c)), HTS-RL synchronizes every $\alpha$ steps. As we show in Sec. 4.2, batch synchronization largely reduces the actor idle time, particularly in environments where the time to perform a step varies.

Decentralized distributed PPO (DD-PPO) [34] is concurrent work which also advocates for synchronous RL. DD-PPO adopts standard synchronous training with a 'preemption threshold,' *i.e.*, once a pre-specified percentage of actors finished rollout, slower actors are terminated. DD-PPO performs distributed training at a large scale, *e.g.*, 128 GPUs, and achieves impressive throughput. However, in each machine DD-PPO follows conventional RL, *i.e.*, alternating between learning and data collection. Also note that the 'preemption threshold' introduces non-determinism. In contrast, HTS-RL targets parallel computing in a single machine. We parallelize learning and data rollout, reduce idle time of actors, and maintain full determinism. HTS-RL can be combined with across machine DD-PPO training which we leave to future work.

## 3 Background

We first introduce notation and Markov Decision Processes (MDPs), review policy gradient and actor-critic algorithms used by A2C [6] and IMPALA [7] and finally discuss the stale-policy issue.

**Reinforcement Learning:** An agent interacts with an environment, collecting rewards over discrete time, as formalized by a Markov Decision Process (MDP). Formally, an MDP $\mathcal{M}(\mathcal{S}, \mathcal{A}, \mathcal{T}, r, H, \gamma)$ is defined by a set of states $\mathcal{S}$, a set of actions $\mathcal{A}$, a transition function $\mathcal{T} : \mathcal{S} \times \mathcal{A} \to \mathcal{S}$ which maps an action and the current agent-state to the next state, a reward function $r : \mathcal{S} \times \mathcal{A} \to \mathbb{R}$, which maps an action and a state to a scalar, a horizon $H$, and a discount factor $\gamma \in (0, 1]$. Note $\mathcal{T}$ and $r$ can be stochastic or deterministic. At each time $t$, the agent selects an action $a_t \in \mathcal{A}$ according to a policy

$\pi$. This policy $\pi$ maps the current state $s_t \in \mathcal{S}$ to a probability distribution over the action space $\mathcal{A}$. Formally, we refer to the output distribution given a state $s_t$ as $\pi(\cdot|s_t)$, and denote the probability of a specific action $a_t$ given a state $s_t$ via $\pi(a_t|s_t)$. After executing the selected action $a_t$, the agent finds itself in state $s_{t+1} = \mathcal{T}(s_t, a_t) \in \mathcal{S}$ and obtains a scalar reward $r_t = r(s_t, a_t)$. The discounted return from a state $s_t$ is $R_t = \sum_{i=0}^{H} \gamma^i r(s_{t+i}, a_{t+i})$. The goal is to find a policy $\pi_\theta$, parameterized by $\theta$, which maximizes the expected discounted return $J(\theta) = \mathbb{E}_{s_t \sim \rho^{\pi_\theta}, a_t \sim \pi_\theta}[R_1]$, where $\rho^{\pi_\theta}$ is the state visitation distribution under $\pi_\theta$.

**Policy Gradient:** To maximize the expected discounted return $J(\theta)$ w.r.t. the parameters $\theta$ of the policy we use its gradient [31] given by

$$\nabla_\theta J(\theta) = \mathbb{E}_{s_t \sim \rho^{\pi_\theta}, a_t \sim \pi_\theta}[\nabla_\theta \log \pi_\theta(a_t|s_t) Q^{\pi_\theta}(s_t, a_t)]. \tag{1}$$

Here $Q^{\pi_\theta}(s_t, a_t) = \mathbb{E}_{s_{i>t} \sim \rho^{\pi_\theta}, a_{i>t} \sim \pi_\theta}[R_t|s_t, a_t]$ is the expected future return obtained by following policy $\pi_\theta$ after action $a_t$ was selected at state $s_t$. Because $Q^{\pi_\theta}$ is unknown, we need to estimate it. For instance, in the REINFORCE algorithm [31], $Q^{\pi_\theta}$ is estimated from $R_t$. To reduce the variance of this estimate, a baseline is subtracted from $R_t$. A popular choice is the value function $V^{\pi_\theta} = \mathbb{E}_{s_{i>t} \sim \rho^{\pi_\theta}, a_i \sim \pi_\theta}[R_t|s_t]$ which estimates the expected future return by following policy $\pi_\theta$ given that we start in state $s_t$. Generally, $V^{\pi_\theta}$ is unknown and therefore often estimated via a function approximator $V_\phi^{\pi_\theta}$ parameterized by $\phi$. Combined we obtain the following expression for the gradient of the expected discounted return:

$$\nabla_\theta J(\theta) = \mathbb{E}_{s_t \sim \rho^{\pi_\theta}, a_t \sim \pi_\theta}[\nabla_\theta \log \pi_\theta(a_t|s_t)(R_t - V_\phi^{\pi_\theta}(s_t))]. \tag{2}$$

Note, it is common to learn the parameters $\phi$ of the value function $V_\phi^{\pi_\theta}$ by minimizing the squared loss $J_V(\phi) = \mathbb{E}_{s_t \sim \rho^{\pi_\theta}, a_t \sim \pi_\theta}[(R_t - V_\phi^{\pi_\theta})^2]$.

**Actor-Critic Algorithms:** In actor-critic algorithms [25, 30], the $n$-step truncated return $R_t^{(n)} = \sum_{i=0}^{n-1} \gamma^i r_{t+i} + \gamma^n V_\phi^{\pi_\theta}(s_{t+k})$ is used to estimate $Q^{\pi_\theta}$. In addition, to encourage exploration, an entropy term $H$ is sometimes added. The gradient of the expected discounted return w.r.t. the policy parameters $\theta$ is

$$\nabla_\theta J(\theta) = \mathbb{E}_{s_t \sim \rho^{\pi_\theta}, a_t \sim \pi_\theta}[\nabla_\theta \log \pi_\theta(a_t|s_t)(R_t^{(n)} - V_\phi^{\pi_\theta}(s_t)) + \lambda H(\pi_\theta(\cdot|s_t))], \tag{3}$$

where $\lambda \geq 0$ controls the strength of the entropy. In practice, the expectation is estimated via

$$\nabla_\theta \hat{J}(\theta, \mathcal{D}^\theta) = \frac{1}{|\mathcal{D}^\theta|} \sum_{(s_t, a_t, r_t) \in \mathcal{D}^\theta} [\nabla_\theta \log \pi_\theta(a_t|s_t)(R_t^{(n)} - V_\phi^{\pi_\theta}(s_t)) + \alpha H(\pi_\theta(\cdot|s_t))], \tag{4}$$

where the set $\mathcal{D}^\theta = \{(s_t, a_t, r_t)\}_t$ subsumes the rollouts collected by following the behavior policy $\pi_\theta$ which is identical to the target policy, *i.e.*, the policy specified by the current parameters.

**Stale Policy Issue:** In asynchronous methods like IMPALA and GA3C, the behavior policy often lags the target policy. Specifically, in IMPALA and GA3C, actors send data to a non-blocking data queue, and learners consume data from the queue as illustrated in Fig. 1(b, c). The data in the queue is stale when being consumed by the learner and delays are not deterministic. Consequently, on-policy algorithms like actor-critic become off-policy. Formally, suppose the latency is $k$, and we refer to the parameters after the $j$-th update as $\theta_j$. Instead of directly estimating the gradient using samples from the current target policy as indicated in Eq. (4), the gradient employed in asynchronous methods is

$$\nabla_{\theta_j} \hat{J}(\theta_j, \mathcal{D}^{\theta_{j-k}}) = \frac{1}{|\mathcal{D}^{\theta_{j-k}}|} \sum_{(s_t, a_t, r_t) \in \mathcal{D}^{\theta_{j-k}}} [\nabla_{\theta_j} \log \pi_{\theta_j}(a_t|s_t) \cdot (R_t^{(n)} - V_\phi^{\pi_{\theta_j}}(s_t)) + \alpha H(\pi_{\theta_j}(\cdot|s_t))]. \tag{5}$$

Note, the gradient w.r.t. $\theta_j$ is estimated using stale data $\mathcal{D}^{\theta_{j-k}}$. This is concerning because the current distribution $\pi_{\theta_j}(a_t|s_t)$ often assigns small probabilities to actions that were taken by the stale policy $\pi_{\theta_{j-k}}$. As a result, $\nabla_{\theta_j} \log \pi_{\theta_j}(a_t|s_t)$ tends to infinity, which makes training unstable and harms the performance as also reported by Babaeizadeh et al. [1] and Espeholt et al. [7].

## 4  High-Throughput Synchronous RL

### 4.1  Method

We aim for the following four features: (1) batch synchronization which reduces actor idle time, (2) learning and rollout take place concurrently which increases throughput, (3) guaranteed lag of only

*one* step between the behavior and target policy which ensures stability of training, (4) asynchronous interaction between actors and executors at the rollout phase to increase throughput while ensuring determinism. In the following we provide an overview before analyzing each of the four features.

**Overview of HTS-RL:** As illustrated in Fig. 1(e), HTS-RL decomposes RL training into executors, actors, and learners. In addition, we introduce two buffers, *i.e.*, the action buffer and the state buffer, as well as two data storages, one for data writing and another one for data reading.

As shown in Fig. 1(e), executors *asynchronously* grab actions from the action buffer, which stores actions predicted by actors as well as a pointer to the environment. Then, executors apply the predicted action to the corresponding environment and observe the next state as well as the reward. The executors then store the received state and an environment pointer within the state buffer, from which actors grab this information *asynchronously*. Next, the actors use the grabbed states to predict the corresponding subsequent actions and send those together with the environment pointer back to the action buffer. The executors also write received data (state, actions, rewards) to one of two storages, *e.g.*, 'storage 2.' Meanwhile, the learners read data from the other data storage, *i.e.*, 'storage 1.'

Note, in HTS-RL, learning and data collection take place concurrently. As shown in Fig. 2(d), the executor and actor processes operate at the same time as the 'learner' process. When executors fill up one of the data storages, *e.g.*, 'storage 2,' learners concurrently consume the data from the other, *e.g.*, 'storage 1.' Eventually the role of the two data storages switches, *i.e.*, the learners consume the data just collected by the executors into 'storage 2,' and the executors start to fill 'storage 1.'

The system does not switch the role of a data storage until executors fill up and learners exhaust the data storage. This synchronization design leads to some idle time for either learner or executor. However, the synchronization is critical to avoid the stale policy issue and to maintain training stability and data efficiency. As shown in Sec. 5, even with this synchronization, HTS-RL has competitive throughput but, importantly, much better data efficiency than asynchronous methods like IMPALA. Given this design, we now discuss the aforementioned four key features in detail.

**Batch synchronization:** To adapt to environments with large step time variance, we deploy batch synchronization in HTS-RL. Specifically, HTS-RL synchronizes actors and executors every $\alpha$ steps. This is indicated via the black vertical lines in Fig. 2(d) which shows an $\alpha = 4$ configuration.

In contrast, A2C (Fig. 2(a)) synchronizes every timestep ($\alpha = 1$), *i.e.*, at every time A2C has to wait for the slowest process to finish. As a result, the throughput of A2C drops significantly in environments with large step time variance. For a detailed analysis on how the step time variance and synchronization frequency impact throughput, please see Sec. 4.2.

**Concurrent rollout and learning:** To enhance throughput for HTS-RL, we ensure that rollout and learning happen concurrently and synchronize every $\alpha$ steps as mentioned before. More specifically, after a synchronization, each executor performs $\alpha$ steps and each learner performs one or more forward and backward passes.

Concurrency contrasts HTS-RL from A2C (Fig. 2(a)) where rollout and learning take place alternatingly. We hence obtain a higher throughput. Note, asynchronous methods such as IMPALA (Fig. 2(b)) also parallelize learning and rollout. However, the intrinsic stale policy issue hurts their performance even when correction methods are applied.

**Delayed gradient:** Due to the two data storages we ensure that the model from the previous iteration is used to collect the data. Staleness is consequently bounded by one step. Different from asynchronous methods such as IMPALA and GA3C, where the delay between target and behavior policy increases with the number of actors and is sensitive to system configurations [1], HTS-RL guarantees that the delay is fixed to one, *i.e.*, the executors are using the parameters of the policy one step prior to synchronization. Please see Sec. 4.2 for a more detailed analysis.

To avoid a stale policy, we use the behavior policy to compute a 'delayed' gradient. We then apply this gradient to the parameters of the target policy. Formally, the update rule of HTS-RL is

$$\theta_{j+1} = \theta_j + \eta \nabla_{\theta_{j-1}} \hat{J}(\theta_{j-1}, \mathcal{D}^{\theta_{j-1}}), \tag{6}$$

where $\eta$ is the learning rate. Note that the gradient $\nabla_{\theta_{j-1}} \hat{J}(\theta_{j-1}, \mathcal{D}^{\theta_{j-1}})$ is computed at $\theta_{j-1}$ before being added to $\theta_j$ ('one-step-delay'). Because the gradient is computed on $\mathcal{D}^{\theta_{j-1}}$ w.r.t. $\theta_{j-1}$, the stale policy issue described in Eq. (5) no longer exists. Using classical results and assumptions, the

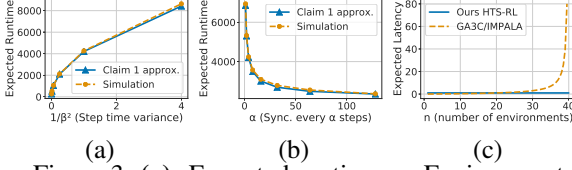
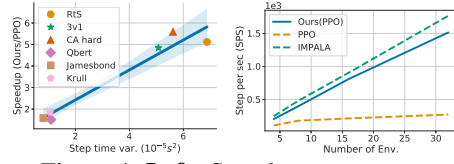

(a)          (b)          (c)

Figure 3: **(a):** Expected runtime *vs.* Environment step time variance ($\frac{1}{\beta^2}$). $\alpha$ is fixed at 4. **(b):** Expected runtime *vs.* synchronization interval ($\alpha$). $\beta$ is fixed at 2. **(c):** Expected latency between behavior policy and target policy *vs.* number of environments ($n$).

Figure 4: **Left:** Speedup *vs.* env. step time variance. RTS: 'Run to score', 3v1: '3 *vs.* 1 with keeper', CA hard: 'Conner attack hard'. **Right:** Step per second (SPS) *vs.* # of env. in GFootball 'counterattack hard.'

convergence rate of the one-step-delayed gradient has been shown to be $O(\frac{1}{\sqrt{T}})$, where $T$ is the number of performed gradient ascent updates [17]. Note that the $O(\frac{1}{\sqrt{T}})$ convergence rate is as good as the zero-delayed case. Please see the appendix for more details and assumptions.

**Asynchronous actors and executors:** To enhance an actor's GPU utilization, in HTS-RL, there are usually fewer actors than executors. More importantly, the actors and executors interact in an asynchronous manner. Specifically, as shown in Fig. 1(e) and Fig. 2(d), actors keep polling an observation buffer for available observations, perform a forward pass for all available observations at once and send the predicted actions back to the corresponding action buffer. This asynchronous interaction prevents actors from waiting for the executors, and thus increases throughput.

However, this asynchrony makes the generated action nondeterministic as actors' sample from a distribution over actions to facilitate exploration. Specifically, with actors grabbing observations asynchronously we can no longer guarantee that a particular actor handles a specific observation. Consequently, even when using pseudo-random numbers within actors, asynchrony will result in nondeterminism. To solve this nondeterminism issue, we defer all randomness to the executors. In practice, along with each observation, an executor sends a pseudo-random number to the observation buffer. The pseudo-random number serves as the random seed for the actor to perform sampling on this observation. Because generation of the pseudo-random numbers in executors does not involve any asynchrony, the predicted actions are deterministic once the random seed for the executor is properly set. Thanks to this, HTS-RL maintains full determinism.

## 4.2 Analysis

As discussed in Sec. 4.1, the run-time is impacted by an environment's step time variance and an RL framework's synchronization frequency. Intuitively, less synchronization and lower step time variance leads to shorter runtime. We will prove this intuition more formally. To this end we let $T_{\text{total}}^{n,K}$ denote the run-time when using $n$ environments to collect $K$ states. We can compute the expected run time $\mathbb{E}[T_{\text{total}}^{n,K}]$ as follows.

**Claim 1.** *Consider collecting $K$ states using $n$ parallel environments. Let $X_i^{(j)}$ denote the time for the $j^{th}$ environment to perform its $i^{th}$ step. Suppose $X_i^{(j)}$ is independent and identically distributed (i.i.d.) and $\sum_{i=1}^{\alpha} X_i^{(j)}$ follows a Gamma distribution with shape $\alpha$ and rate $\beta$. Assume the computation time of each actor consistently takes time $c$. Given these assumptions, the expected time $\mathbb{E}[T_{total}^{n,K}]$ to generate $K$ states approaches*

$$\frac{K}{n\alpha}\left(\frac{\gamma}{\beta}\left(1 + \frac{\alpha - 1}{\beta F^{-1}(1 - \frac{1}{n})}\right) + F^{-1}(1 - \frac{1}{n})\right) + \frac{Kc}{n}, \tag{7}$$

*where $F^{-1}$ is the inverse cumulative distribution function (inverse CDF) of a gamma distribution with shape $\alpha$ and rate $\beta$,* i.e.*, Gamma$(\alpha, \beta)$, and $\gamma$ is the Euler-Mascheroni constant.*

*Proof.* See supplementary material. □

We perform a simulation to verify the tightness of the derived expected runtime. As shown in Fig. 3(a,b), where we assume the step time follows an exponential distribution with rate $\beta$, *i.e.*, $\exp(\beta)$, the derived approximation accurately fits the simulation results. Note that the sum of $\alpha$ i.i.d. exponential random variables ($\exp(\beta)$) is a Gamma$(\alpha, \beta)$ random variable. Fig. 3(a) shows that for

a fixed synchronization interval ($\alpha = 4$), the total runtime increases rapidly if the variance of the environment step time increases. To reduce runtime, we can perform batch synchronization, *i.e.*, $\alpha > 1$. As shown in Fig. 3(b) for a rate $\beta = 2$, the total runtime decreases when we increase the synchronization interval $\alpha$. Note, when $\alpha \geq \frac{K}{n}$, Eq. (7) returns the time of an asynchronous system.

While fully asynchronous systems enjoy shorter runtime, the stale policy issue worsens when the number of actors increases. Formally, we let $L$ denote the latency between behavior policy and target policy in GA3C and IMPALA. We obtain the following result for the expected latency $\mathbb{E}[L]$.

**Claim 2.** *Consider asynchronous parallel actor-learner systems, such as GA3C and IMPALA. Suppose the system has $n$ actors and each actor is sending data to the data queue following an i.i.d. Poisson distribution with rate $\lambda_0$. The learners consume data following an exponential distribution with rate $\mu$. Let $L$ denote the latency between the behavior policy and the target policy. Then, we have $\mathbb{E}[L] = \frac{n\rho_0}{1-n\rho_0}$, where utilization $\rho_0 = \frac{\lambda_0}{\mu}$.*
*Proof.* See supplementary material. □

In the GFootball environment, an actor generates about $\lambda_0 = 100$ frames per second while a reasonable learner consumes $\mu = 4000$ frames per second. The expected latency for this setting is shown in Fig. 3(c). With few actors, the latency is small. However, this means the learners are mostly idle. Note that the latency grows rapidly when the number of actors grow. This large policy lag leads to unstable training. In contrast, the latency of HTS-RL is always one regardless of the number of deployed actors. As a result, HTS-RL enjoys stable training and good data efficiency.

## 5  Experiments

**Environments.** We evaluate the proposed approach on a subset of the Atari games [2, 3] and all 11 academy scenarios of the recently introduced Google Research Football (GFootball) [15] environment. For the Atari environment we use raw images as agent input. For GFootball, the input is either raw images or an 'extracted map' which is a simplified spatial representation of the current game state.

**Experimental Setup.** We compare the proposed approach with IMPALA, A2C, and PPO. For IMPALA, we use the Torch Beast implementation [16], which reports better performance than the official IMPALA implementation [7]. For A2C and PPO, we use the popular synchronous Pytorch implementation of Kostrikov [14].  For both Atari and GFootball experiments, we use the same models as Küttler et al. [16] and Kurach et al. [15]. We follow their default hyperparameter settings. Please see the supplementary material for details on hyperparameter settings and model architectures. Note, we study parallel RL training on a single machine while IMPALA is a distributed training algorithm, which performs RL training across multiple machines. For a fair comparison, we use 16 parallel environments on a single machine for all methods. Importantly, while being downscaled to one machine, the reported IMPALA results match the results reported in the original papers [15, 16]. Following Kostrikov [14], all methods are trained for 20M environment steps in the Atari environment. For GFootball, following Kurach et al. [15], we use 5M steps.

**Evaluation protocol.** To ensure a rigorous and fair evaluation, Colas et al. [4], Henderson et al. [11] suggest reporting the '*final metric*.' The *final metric* is an average over the last 100 evaluation episodes, *i.e.*, 10 episodes for each of the last ten policies during training. However, Colas et al. [4], Henderson et al. [11] don't consider runtime measurements as a metric. To address this, we introduce the '*final time metric*' and the '*required time metric*.' The '*final time metric*' reports the *final metric* given a limited training time. Training is terminated when the time limit is reached, and the final metric at this point of time is the '*final time metric*.' The *required time metric* reports the runtime required to achieve a desired average evaluation episode reward. The average evaluation reward is the running average of the most recent 100 evaluation episodes, *i.e.*, 10 episodes for each of the most recent ten policies. All experiments are repeated for five runs with different random seeds. All the plots in Fig. 5 include mean of the five runs and the $95\%$ confidence interval obtained by using the Facebook Bootstrapped implementation with 10,000 bootstrap samples. For Atari experiments, we follow the conventional 'no-op' procedure, *i.e.*, at the beginning of each evaluation episode, the agents perform up to 30 'no-op' actions.

**Results.** To confirm that Claim 1 holds for complex environments such as Atari and GFootball, we study the speedup ratio of HTS-RL to A2C/PPO baselines when the step time variance changes. As shown in Fig. 4(left), in environments with small variance, HTS-RL is around $1.5\times$ faster than the baselines. In contrast, in environments with large step time variance, HTS-RL is more than $5\times$

| Method | IMPALA | A2C | Ours (A2C) |
|---|---|---|---|
| BankHeist | $339 \pm 10$ | $775 \pm 166$ | $\mathbf{942 \pm 100}$ |
| Beam Rider | $4000 \pm 690$ | $4392 \pm 134$ | $\mathbf{6995 \pm 420}$ |
| Breakout | $201 \pm 133$ | $362 \pm 29$ | $\mathbf{413 \pm 37}$ |
| Frostbite | $73 \pm 2$ | $272 \pm 14$ | $\mathbf{315 \pm 12}$ |
| Jamesbond | $82 \pm 10$ | $438 \pm 59$ | $\mathbf{474 \pm 88}$ |
| Krull | $2546 \pm 551$ | $7560 \pm 892$ | $\mathbf{7737 \pm 609}$ |
| KFMaster | $9516 \pm 3311$ | $\mathbf{30752 \pm 6641}$ | $30020 \pm 3559$ |
| MsPacman | $807 \pm 170$ | $1236 \pm 292$ | $\mathbf{1675 \pm 459}$ |
| Qbert | $4116 \pm 610$ | $12479 \pm 1965$ | $\mathbf{13682 \pm 1873}$ |
| Seaquest | $458 \pm 2$ | $\mathbf{1833 \pm 6}$ | $1831 \pm 7$ |
| S. Invader | $\mathbf{1142 \pm 207}$ | $596 \pm 69$ | $731 \pm 80$ |
| Star Gunner | $8560 \pm 918$ | $41414 \pm 3826$ | $\mathbf{52666 \pm 5182}$ |

Table 1: Atari experiment in *final time metrics*: Average evaluation rewards achieved given limited training time.

| Method | IMPALA | PPO | Ours (PPO) |
|---|---|---|---|
| Empty goal close | 1.7/2.6 | 5.4/15.5 | **1.0/2.0** |
| Empty goal | 8.4/11.7 | 12.8/19.2 | **2.0/3.9** |
| Run to score | 27.0/34.6 | 16.2/32.5 | **6.3/11.4** |
| RSK | 52.3/- | 51.2/68.2 | **11.5/18.8** |
| PSK | -/- | 70.0/- | **38.8/-** |
| RPSK | 22.3/**25.4** | 45.2/90.8 | **13.5**/27.1 |
| 3 *vs*. 1 w/ keeper | -/- | 67.4/144.2 | **15.9/25.6** |
| Corner | -/- | -/- | -/- |
| Counterattack easy | -/- | 223.2/- | **91.3/-** |
| Counterattack hard | -/- | 383.4/- | **61.8/-** |
| 11 vs 11 w/ lazy Opp. | 58.2/- | 95.8/260.9 | **14.4/72.1** |

Table 2: GFootball results in *required time metrics:* required time (minutes) to achieve scores (time to achieve score 0.4 / time to achieve score 0.8).

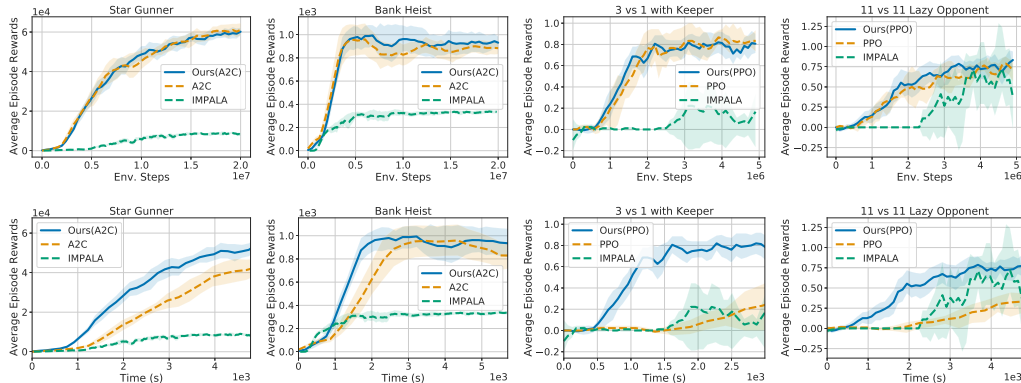

Figure 5: Training curves. **Top row:** reward *vs*. environment steps. **Bottom row:** reward *vs*. time.

faster. To ensure the scalability of HTS-RL, we investigate how the throughput, *i.e.*, the steps per second (SPS), changes when the number of environments increases. For this we conduct experiments on the GFootball 'counterattack hard' scenario, which has the longest step time and largest step time variance among all GFootball scenarios. As shown in Fig. 4(right), the throughput of HTS-RL with PPO increases almost linearly when the number of environments increases. In contrast, for the synchronous PPO baseline, the throughput increase is marginal.

To study throughput and data efficiency on rewards, we speed up A2C with HTS-RL, and compare to baseline A2C and IMPALA on Atari games. Tab. 1 summarizes the *final time metric* results. The time limit for all experiments is set to the time when the fastest method, IMPALA, finishes training, *i.e.*, when it reached the 20M environment step limit. As shown in Tab. 1, given the same amount of time, HTS-RL with A2C achieves significantly higher rewards than IMPALA and A2C baselines.

We also speedup PPO with HTS-RL and compare with baseline PPO and IMPALA on GFootball environments. Tab. 2 summarizes the results using the *required time metric*. We reported the required time to achieve an average score of 0.4 and 0.8. Note that because each episode terminates when one of the teams scores, the maximum possible reward for GFootball scenarios is 1.0. The results are summarized in Tab. 2, where '-' indicates that the method did not achieve an average score of 0.4 and 0.8 after 5M environment step training. As shown in Tab. 2, our HTS-RL achieves the target score, *i.e.*, 0.4 and 0.8 significantly faster than the PPO and IMPALA baselines. Please see the supplementary material for more comparisons, all training curves, and results of final metric and required time metric. Fig. 5 shows the training curves for both Atari and Gfootball experiments. The first row of Fig. 5 shows that HTS-RL achieves similar data efficiency than baseline synchronous methods. In contrast, asynchronous baseline IMPALA has much lower data efficiency. The second row of Fig. 5 shows that HTS-RL achieves higher reward than A2C, PPO and IMPALA in shorter training time.

Thanks to high throughput and stability, we can demo HTS-RL's potential on more complex RL tasks. Specifically, we scale HTS-RL to training of multiple players in the '3 *vs*. 1 with keeper' scenario of

|  | 1 Agent (Raw Image) | 3 Agents (Raw Image) |
|---|---|---|
| Ours (PPO) | 0.30± 0.11 | **0.63± 0.15** |

Table 3: Average game score of 8M step multi-agent training with raw image input on '3 *vs.* 1 with keeper' (*final metrics*).

| Number of Actors | 1 | 4 | 8 | 16 |
|---|---|---|---|---|
| Steps per second (SPS) | 1229 | 1362 | 1393 | 1388 |
| Average scores | 0.82 | 0.82 | 0.82 | 0.82 |

Table 4: Different number of actors on '3 *vs.* 1 with keeper'. Average scores: average score of 100 evaluation episodes.

GFootball from only raw image input. To the best of our knowledge, this is the first work to train multiple agents *from only raw image input* in this environment. The results are summarized in Tab. 3. Training on three agents achieves higher scores than training on a single agent. Note, we only use one machine with 4 GPUs.

**Ablation study.** To investigate how the number of actors impacts the performance of HTS-RL, we run HTS-RL with 1, 4, 8, and 16 actors on '3 *vs.* 1 with keeper'. The results are summarized in Tab. 4, where 'average scores' gives the average score of 100 evaluation episodes. The SPS improvement when using more than 4 actors is marginal, because the GFootball game engine process time dominates. That is, when using more than four actors, most of the time, actors are waiting for response from the GFootball game engine. Note that thanks to full determinism of HTS-RL, different actor numbers have identical final average scores.

To study the synchronization interval's impact on the performance of HTS-RL, we use synchronization intervals 4, 16, 64, 128, 256, 512 on the '3 *vs.* 1 with keeper' task. As shown in Tab. 5, longer synchronization intervals gives higher throughput, which further confirms Claim 1 and simulation results in Fig. 3(b). Note HTS-RL achieves consistent high average scores while the synchronization interval changes.

## 6 Conclusion

We develop High-Throughput Synchronous RL (HTS-RL). It achieves a high throughput while maintaining data efficiency. For this HTS-RL performs batch synchronization, and concurrent rollout and learning. Moreover, HTS-RL avoids the 'stale-policy' issue.

**Acknowledgements:**

This work is supported in part by NSF under Grant # 1718221, 2008387 and MRI #1725729, NIFA award 2020-67021-32799, UIUC, Samsung, Amazon, 3M, Cisco Systems Inc. (Gift Award CG 1377144), and a Google PhD Fellowship to RY. We thank Cisco for access to the Arcetri cluster.

| Synchronization Interval | 4 | 16 | 64 | 128 | 256 | 512 |
|---|---|---|---|---|---|---|
| Steps per second (SPS) | 445 | 1070 | 1345 | 1349 | 1360 | 1377 |
| Average scores | 0.82 | 0.81 | 0.81 | 0.82 | 0.81 | 0.83 |

Table 5: Different synchronization interval on '3 *vs.* 1 with keeper'. Average scores: average score of 100 evaluation episodes.

## Broader Impact

We think artificial intelligence (AI) algorithms can significantly improve people's lives and should be accessible to everyone. This paper introduces a high-throughput system which permits efficient deep RL training on a single machine. We believe efficient training of deep RL with limited computational resources is critical to make artificial intelligence more accessible, particularly for people and institutions who can not afford a costly distributed system.

In the past decade, deep RL achieved impressive results on complex tasks such as GO and 3D video games. However, those achievements rely on a huge amount of computing resources, *e.g.*, AlphaGo [28] requires 1920 CPUs and 280 GPUs. In contrast, regular personal desktops have only 4 to 16 CPUs and no more than 4 GPUs. As a result, deep RL's huge demand for computational resources makes it a research privilege for large companies and institutions which can afford these resources.

This paper serves as a step toward making deep RL AI accessible to everyone. With the proposed HTS-RL it is possible to train deep RL agents on 3D video games using a regular desktop with 16 CPUs and 4 GPUs instead of requiring a costly distributed system.

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
