[Supplementary Material]

# Appendix: High-Throughput Synchronous Deep RL

In this appendix we first provide the proofs for Claim 1 (Sec. A) and Claim 2 (Sec. B). We then discuss delayed gradient updates (Sec. C), additional ablation studies (Sec. D), comparison with additional baselines(Sec. E), implementation details (Sec. F), metrics (Sec. G) and provide all the training curves (Sec. H).

## A    Proof of Claim 1

**Claim 1.** *Consider collecting $K$ states using $n$ parallel environments. Let $X_i^{(j)}$ denote the time for the $j^{th}$ environment to perform its $i^{th}$ step. Suppose $X_i^{(j)}$ is independent and identically distributed (i.i.d.) and $\sum_{i=1}^{\alpha} X_i^{(j)}$ follows a Gamma distribution with shape $\alpha$ and rate $\beta$. Assume the computation time of each actor consistently takes time $c$. Given these assumptions, the expected time $\mathbb{E}[T_{total}^{n,K}]$ to generate $K$ states approaches*

$$\frac{K}{n\alpha}\left(\frac{\gamma}{\beta}\left(1 + \frac{\alpha-1}{\beta F^{-1}(1-\frac{1}{n})}\right) + F^{-1}\left(1 - \frac{1}{n}\right)\right) + \frac{Kc}{n}, \tag{7}$$

*where $F^{-1}$ is the inverse cumulative distribution function (inverse CDF) of a gamma distribution with shape $\alpha$ and rate $\beta$, i.e., Gamma$(\alpha, \beta)$, and $\gamma$ is the Euler-Mascheroni constant.*

*Proof.* As the environments synchronize every $\alpha$ steps, we need $\frac{K}{n\alpha}$ synchronizations to finish the $K$ steps. Let $T_l$ denote the time required for the $l^{\text{th}}$ synchronization. We have $\mathbb{E}[T_{\text{total}}^{n,K}] = \sum_{l=1}^{\frac{K}{\alpha n}} \mathbb{E}[T_l]$. Note $\mathbb{E}[T_l] = \mathbb{E}[\max_j \sum_{i=1}^{\alpha} X_i^{(j)}] + \alpha c \ \forall l$. By assumption we know that $Y_j \triangleq \sum_{i=1}^{\alpha} X_i^{(j)}$ follows a gamma distribution with shape $\alpha$ and rate $\beta$. By extreme value theory [5, 9], suppose $X_l^{(j)} \sim$ Gamma$(\alpha, \beta)$, then $\mathbb{E}[\max_j X_l^{(j)}] \simeq \frac{\gamma}{\beta}(1 + \frac{\alpha-1}{\beta F^{-1}(1-\frac{1}{n})}) + F^{-1}(1 - \frac{1}{n})$, where $F^{-1} = \inf\{x \in \mathcal{R} : F(x) \geq q\}$. $F(x)$ is the CDF of Gamma$(\alpha, \beta)$, and $\gamma$ is the Euler-Mascheroni constant. By plugging the obtained approximation into $\sum_{l=1}^{\frac{K}{\alpha n}} \mathbb{E}[T_l]$, the result follows.     $\square$

In Claim 1, we assume the sum of steptimes (synchronization time) to follow a Gamma distribution. We empirically verify this assumption. In Fig. A1, we show the histogram of synchronization time (sum of every 100 step times) on '3 *vs*. 1 w/ keeper' Furthermore, we perform a Kolmogorov-Smirnov goodness-of-fit test, with a significance-level of 0.05 and D-statistics of 0.04. We find the empirical data is consistent with the assumed Gamma distribution.

## B    Proof of Claim 2

**Claim 2.** *Consider asynchronous parallel actor-learner systems, such as GA3C and IMPALA. Suppose the system has $n$ actors and each actor is sending data to the data queue following an i.i.d. Poisson distribution with rate $\lambda_0$. The learners consume data following an exponential distribution*

Figure A1: Empirical synchronization time.

with rate $\mu$. Let $L$ denote the latency between the behavior policy and the target policy. Then, we have $\mathbb{E}[L] = \frac{n\rho_0}{1-n\rho_0}$, where utilization $\rho_0 = \frac{\lambda_0}{\mu}$.

*Proof.* Observe that the latency $L$ is equal to the length of the data queue. $n$ actors send data to the queue with rate $n\lambda_0$ in total. Let $P_i$ denote the probability that there are $i$ data points in the queue. To be stable, the system must satisfy the balance equations

$$n\lambda_0 P_0 = \mu P_1 \tag{A1}$$

$$(n\lambda_0 + \mu)P_j = n\lambda_0 P_{j-1} + \mu P_{j+1}, \qquad j \geq 1. \tag{A2}$$

Note Eq. (A1) and Eq. (A2) reduce to

$$(n\lambda_0)P_j = \mu P_{j+1}, j \geq 0, \tag{A3}$$

or $P_{j+1} = n\rho_0 P_j, j \geq 0$ from which we recursively obtain

$$P_j = (n\rho_0)^j P_0. \tag{A4}$$

Using the fact that $1 = \sum_{j=0}^{\infty} P_j = P_0 \sum_{j=0}^{\infty} (n\rho_0)^j$, we observed that there is a solution if and only if $n\rho_0 < 1$, in which case $1 = P_0(1 - n\rho_0)^{-1}$, or

$$P_0 = 1 - n\rho_0. \tag{A5}$$

Therefore, we have

$$P_j = (n\rho_0)^j (1 - n\rho_0), \tag{A6}$$

which follows a geometric distribution with success probability $(1 - n\rho_0)$. Therefore, we have $\mathbb{E}[L] = \frac{n\rho_0}{1-n\rho_0}$ which concludes the proof. □

## C   Delayed Gradient

A delayed stochastic gradient descent performs the following update: $\theta_t = \theta_{t-1} - \alpha_t \nabla \ell(x_{t-\tau}; \theta_{t-\tau})$. The algorithm is identical to the standard stochastic gradient descent, except that gradients are delayed by a time step of $\tau$.

Consider a loss function of the form

$$\mathcal{L}(\theta) \triangleq \sum_{t=1}^{T} \ell(x_t; \theta). \tag{A7}$$

We are interested in analyzing the convergence rate of $\theta$ to the optimal parameters $\theta^* \triangleq \arg\min_\theta \mathcal{L}(x_t; \theta)$. Following Langford et al. [17], we assume the following: (a) $\ell$ is convex, (b) $L$-Lipschitz, *i.e.*, $\|\nabla(\ell(x, \theta))\| \leq L$, (c) $x_t$ is drawn i.i.d. following a uniform distribution from a finite set $X$, (d) $\max_{x,x' \in X} \frac{1}{2}\|x - x'\|^2 \leq F^2$, where $F$ is a constant, and (e) the learning rate of the delayed stochastic gradient descent is $\frac{\sigma}{\sqrt{t-\tau}}$, where $\sigma^2 = \frac{F^2}{2\tau L^2}$, then

$$\sum_{t=1}^{T} \ell(x_t, \theta_t) - \ell(x_t, \theta^*) \leq 4FL\sqrt{\tau T}. \tag{A8}$$

Dividing both sides by $T$, we have

$$\frac{1}{T} \sum_{t=1}^{T} \ell(x_t, \theta_t) - \ell(x_t, \theta^*) \leq 4FL\sqrt{\frac{\tau}{T}}. \tag{A9}$$

Stated differently, the convergence rate is $O(\sqrt{\frac{\tau}{T}})$. For HTS-RL with on-policy RL algorithms, the delay is guaranteed to be one, *i.e.*, $\tau = 1$. Therefore, the convergence rate is $O(\sqrt{\frac{1}{T}})$. Note that in practice the aforementioned assumptions are typically not met due to the use of deep nets.

## D   Ablation Study

|  | Our Delayed Gradient | Truncated I.S. | No Correction |
|---|---|---|---|
| BankHeist | **987** | 881 | 877 |
| Breakout | **415** | 390 | 402 |
| Seaquest | **1831** | 1827 | 1784 |

Table A1: Average episode rewards of our delayed-gradient, truncated importance sampling, and no correction on Atari games.

## D.1 Delayed Gradient

In addition to the convergence rate bound of delayed gradient, we verify the effectiveness of delayed gradient empirically. We run HTS-RL with (1) delayed gradient, (2) truncated importance sampling, and (3) no correction on multiple Atari games. The results are summarized in Tab. A1, where the average rewards of 100 evaluation episodes are reported. Compared with truncated importance sampling and no correction, the one-step delayed gradient in HTS-RL achieves a higher reward, which underlines the suitability of the delayed gradient strategy.

Figure A2: HTS-RL and SeedRL on GFootball 11 *vs.* 11 easy task.

| Method | Kostrikov [14] | OpenAI Baselines [6] | rlpyt [29] | Ours |
|---|---|---|---|---|
| BankHeist | 1382 ±6 | 991 ±14 | 1737 ±39 | **2111 ±21** |
| Beam Rider | 1663 ±14 | 1081 ±18 | 2086 ±32 | **2586 ±14** |
| Breakout | 1225 ±12 | 829 ±31 | 1508 ±60 | **1885 ±15** |
| Frostbite | 1337 ±8 | 962 ±15 | 1803 ±17 | **1973 ±24** |
| Jamesbond | 1353 ±5 | 1014 ±1 | 1991 ±24 | **2139 ±31** |
| Krull | 1443 ±6 | 1057 ±11 | 2001 ±29 | **2657 ±16** |
| KFMaster | 1532 ±15 | 1056 ±8 | 1979 ±55 | **2483 ±15** |
| MsPacman | 1574 ±9 | 1052 ±3 | 1972 ±13 | **2364 ±5** |
| Qbert | 1232 ±13 | 953 ±7 | 1621 ±43 | **1860 ±6** |
| Seaquest | 1593 ±10 | 946 ±21 | 1918 ±25 | **2633 ±32** |
| S. Invader | 1514 ±20 | 1010 ±7 | 1899 ±32 | **2318 ±12** |
| Star Gunner | 1622 ±19 | 1110 ±5 | 2066 ±24 | **2616 ±25** |

Table A2: SPS of different implementations of A2C.

## E Baselines

### E.1 A2C

To ensure the A2C implementation [14] we use is a strong baseline, we compare the speed of different versions of A2C, including our HTS-RL, Kostrikov [14], OpenAI baselines [6], and rlpyt [29], on Atari games. For a fair comparison, all methods use 16 parallel environment processes for data collection, and one GPU for model training/forwarding. For rlpyt, we use the most efficient 'parallel-

|  | IMPALA | A2C/Ours |
|---|---|---|
| Unroll length | 20 | 5 |
| Batch size | 32 | - |
| Discount factor | 0.99 | 0.99 |
| Value loss coefficient | 0.5 | 0.5 |
| Entropy loss coefficient | 0.01 | 0.01 |
| RMSProp momentum | 0.00 | 0.00 |
| RMSProp $\epsilon$ | 0.01 | 0.00001 |
| Learning rate | 0.006 | 0.0007 |
| Number of actors | 16 | 4 |

Table A3: Hyper-parameters of IMPALA and A2C/Ours(A2C) in Atari experiments.

| Method | IMPALA 16 actors (Baseline) | IMPALA 48 actors [16] |
|---|---|---|
| BankHeist | 339 | ∼300 |
| Beam Rider | 4000 | ∼4000 |
| Breakout | 201 | ∼130 |
| Frostbite | 73 | ∼70 |
| Jamesbond | 82 | ∼80 |
| Krull | 2546 | ∼2500 |
| KFMaster | 9516 | ∼8000 |
| MsPacman | 807 | ∼1300 |
| Qbert | 4116 | ∼4000 |
| Seaquest | 458 | ∼420 |
| S. Invader | 1142 | ∼2000 |
| Star Gunner | 8560 | ∼6000 |

Table A4: Atari@20M environment steps. Since Küttler et al. [16] don't report exact scores at 20M environment steps, we obtain their numbers from their plots and indicate that with a ∼ symbol.

GPU' mode. As shown in Tab. A2, Kostrikov's A2C is a strong baseline, which achieves $1.4\times$ higher SPS than OpenAI baselines. Also, HTS-RL consistently achieves higher SPS than rlpyt.

### E.2 SeedRL

SeedRL [8] is a recent work that reports results on GFootball '11 *vs*. 11 easy' task. We compare HTS-RL with Seed RL (V-trace) [8] on Gfootball '11 *vs*. 11 easy.' For a fair comparison, both HTS-RL and Seed RL use 16 parallel environment processes and one GPU. HTS-RL achieves 829 environment steps per second (SPS) while Seed RL achieves 609 SPS. After 20M steps of training, HTS-RL and Seed RL achieve a $3.55 \pm 0.3$ and $1.50 \pm 0.7$ score difference, respectively. The training curve is shown in Fig. A2.

## F Implementation Details

### F.1 Atari Game Experiments

In Atari experiments, we use the same neural network architectures as Espeholt et al. [7], Küttler et al. [16] for all three methods (IMPALA, A2C, Ours). The network has four hidden layers. The first layer is a convolutional layer with 32 filters of size $8 \times 8$ and stride 4. The second layer is a convolutional layer with 64 filters of size $4 \times 4$ and stride 2. The third layer is a convolutional layer with 64 filters of size $3 \times 3$ and stride 1. The fourth layer is a fully connected layer with 512 hidden units. Following the hidden units are two sets of output. One provides a probability distribution over all valid actions. The other one provides the estimated value function. For ours (A2C) and A2C baseline, we use the same hyper-parameters as Kostrikov [14]. For IMPALA, we use the same hyper-parameters as Espeholt et al. [7], Küttler et al. [16]. We summarize the hyper-parameters in Tab. A3. Note Küttler et al. [16] deploy distributed IMLALA with 48 actors. However, in this work we target single machine parallel computing, and restrict ourselves to 16 parallel environments. For a fair comparison, we run all experiments with 16 parallel environments on a single machine. Importantly, while being downscaled to one machine, the reported IMPALA results match the results reported in the original paper [16]. Tab. A4 summarizes the results of our baseline and that reported by Küttler et al. [16].

### F.2 GFootball Experiments

In GFootball experiments, we use the CNN architecture of Kurach et al. [15] for all three methods (IMPALA, PPO, Ours). The network has four hidden layers. The first layer is a convolutional layer with 32 filters of size $8 \times 8$ and stride 4. The second layer is a convolutional layer with 64 filters of

| Method | IMPALA 16 actors (Baseline) | IMPALA 500 actors [15] |
|---|---|---|
| Empty goal close | 1.0 | ∼0.99 |
| Empty goal | 1.0 | ∼0.85 |
| Run to score | 0.80 | ∼ 0.80 |
| RSK | 0.05 | ∼0.22 |
| PSK | 0.20 | ∼0.18 |
| RPSK | 0.82 | ∼0.41 |
| 3 vs 1 w/ keeper | 0.21 | ∼0.20 |
| Corner | 0.0 | ∼-0.1 |
| Counterattack easy | 0.0 | ∼0.0 |
| Counterattack hard | 0.50 | ∼0.0 |
| 11 vs 11 w/ lazy Opp. | 0.71 | ∼0.38 |

Table A5: GFootball Academy@5M environment steps. Since Kurach et al. [15] don't report exact scores at 5M environment steps, we obtain their numbers from their plots and indicate that with a ∼ symbol.

size $4 \times 4$ and stride 2. The third layer is a convolutional layer with 64 filters of size $3 \times 3$ and stride 1. The fourth layer is a fully connected layer with 512 hidden units. Following the hidden units are two sets of output. One provides a probability distribution over all valid actions. The other one provides the estimated value function. To be consistent with the official torch beast implementation [16], we use RMSProp for all methods. Regarding hyper-parameters, for ours (PPO) and PPO baseline, we mostly use the same hyper-parameters as Kurach et al. [15]. The only difference is that, instead of 512 steps, we unroll for 128 steps, which we found to give better results. For IMPALA, Kurach et al. [15] deploy distributed training with 500 actors. However, in this work we target single machine parallel computing, and restrict ourselves to 16 parallel environments. Therefore, for a fair comparison, we mostly follow the hyper-parameter settings of Kurach et al. [15], but decrease the number of actors and batch size. With only 16 actors and a smaller batch size, our baseline results match the results of IMPALA on GFootball environments reported by Kurach et al. [15]. Tab. A5 summarizes the results of our baseline and that reported by Kurach et al. [15]. The hyper-parameters are summarized in Tab. A6.

## G    Final Time and Required Time Metrics

The results of all Atari experiments in *final time metric* and *required time metric* are summarized in Tab. A7 and Tab. A8. For *final time metric*, the time limit for each experiment is set to the time when IMPALA finishes training for 20M steps. For *required time metric*, we report the time to achieve average episode rewards of $40\%$ and $80\%$ of the A2C baseline episode rewards reported by Dhariwal et al. [6]. The results of all GFootball experiments in *final time metric* and *required time metric* are summarized in Tab. A9 and Tab. A10. For *final time metric*, the time limit for each experiment is set to the time when IMPALA finishes training for 5M steps. For *required time metric*, we report the time to achieve an average score of 0.4 and 0.8. As shown in Tab. A7 and Tab. A9, given the same amount of time, HTS-RL consistently achieves higher average rewards/scores than IMPALA and synchronous baselines. Moreover, as shown in Tab. A8 and Tab. A10, to achieve a target reward/score, HTS-RL consistently needs less time.

## H    Training Curves

The training curves of all Atari and GFootball experiments in terms of time and number of environment steps are shown in Fig. A3, Fig. A4, Fig. A5, and Fig. A6. As shown in Fig. A4 and Fig. A6, HTS-RL does not trade data efficiency for higher throughput. While achieving much higher throughput, HTS-RL still maintains a data efficiency similar to synchronous baselines. As a result, HTS-RL consistently achieves higher rewards in shorter time than IMPALA and synchronous baselines across different environments (Fig. A3, Fig. A5).

|  | IMPALA | PPO / PPO(Ours) |
|---|---|---|
| Unroll length | 32 | 128 |
| Batch size | 8 | 16 |
| Discount factor | 0.993 | 0.993 |
| Value loss coefficient | 0.5 | 0.5 |
| Entropy loss coefficient | 0.00087453 | 0.003 |
| RMSProp momentum | 0.00 | 0.00 |
| RMSProp $\epsilon$ | 0.01 | 0.00001 |
| Learning rate | 0.00019896 | 0.000343 |
| Number of actors | 16 | 4 |

Table A6: Hyper-parameters of IMPALA and PPO/Ours(PPO) in GFootball experiments.

| Method | IMPALA | A2C | Ours (A2C) |
|---|---|---|---|
| BankHeist | $339 \pm 10$ | $775 \pm 166$ | $\mathbf{942 \pm 100}$ |
| Beam Rider | $4000 \pm 690$ | $4392 \pm 134$ | $\mathbf{6995 \pm 420}$ |
| Breakout | $201 \pm 133$ | $362 \pm 29$ | $\mathbf{413 \pm 37}$ |
| Frostbite | $73 \pm 2$ | $272 \pm 14$ | $\mathbf{315 \pm 12}$ |
| Jamesbond | $82 \pm 10$ | $438 \pm 59$ | $\mathbf{474 \pm 88}$ |
| Krull | $2546 \pm 551$ | $7560 \pm 892$ | $\mathbf{7737 \pm 609}$ |
| KFMaster | $9516 \pm 3311$ | $\mathbf{30752 \pm 6641}$ | $30020 \pm 3559$ |
| MsPacman | $807 \pm 170$ | $1236 \pm 292$ | $\mathbf{1675 \pm 459}$ |
| Qbert | $4116 \pm 610$ | $12479 \pm 1965$ | $\mathbf{13682 \pm 1873}$ |
| Seaquest | $458 \pm 2$ | $\mathbf{1833 \pm 6}$ | $1831 \pm 7$ |
| S. Invader | $\mathbf{1142 \pm 207}$ | $596 \pm 69$ | $731 \pm 80$ |
| Star Gunner | $8560 \pm 918$ | $41414 \pm 3826$ | $\mathbf{52666 \pm 5182}$ |

Table A7: Atari experiment in *final time metrics*: Average evaluation rewards achieved given limited training time.

| Method (target reward 1 / target reward 2) | IMPALA | A2C | Ours (A2C) |
|---|---|---|---|
| BankHeist (480 / 960) | -/- | 28.9/**62.1** | **18.9**/116.8 |
| Beam Rider (1600 / 3200) | 36.4/60.4 | 32.1/54.5 | **10.3/30.9** |
| Breakout (160 / 320) | 77.8/- | 21.7/43.5 | **17.7/38.9** |
| Frostbite (104 / 208) | -/- | 5.0/10.0 | **3.4/6.8** |
| Jamesbond (200 / 400) | -/- | 39.4/49.2 | **21.8/31.1** |
| Krull (3600 / 7200) | -/- | 13.9/37.0 | **7.5/37.6** |
| KFMaster (15200 / 30420) | -/- | 39.5/192.6 | **18.8/118.1** |
| MsPacman (880 / 1760) | 75.8/- | 49.3/160.3 | **22.9/94.3** |
| Qbert (4000 / 8000) | 94.2/- | 53.3/83.8 | **52.2/67.7** |
| Seaquest (640 / 1280) | -/- | 6.7/28.4 | **4.0/17.2** |
| Space. (240 / 480) | **9.65/19.9** | 14.1/26.4 | 6.9/21.8 |
| Star Gunner (8400 / 16800) | 47.1/- | 28.7/41.1 | **17.8/25.4** |

Table A8: Atari experiment in *required time metrics*: Required time (minutes) to achieve goal episode rewards (time required to achieve 40% rewards reported by Dhariwal et al. [6] / time required to achieve 80% rewards reported by Dhariwal et al. [6]). '-' indicates that the method did not achieve the desired reward after 20M environment step training. Space.: Space invaders, KFMaster: KungFu Master.

| Method | IMPALA | PPO | Ours (PPO) |
|---|---|---|---|
| Empty goal close | **1.00 ± 0.00** | **1.00 ± 0.00** | **1.00 ± 0.00** |
| Empty goal | **1.00 ± 0.00** | 0.89 ±0.05 | 0.94 ±0.04 |
| Run to score | 0.65±0.42 | 0.89±0.05 | **0.93±0.03** |
| RSK | 0.03±0.03 | 0.52±0.21 | **0.88±0.06** |
| PSK | 0.00±0.00 | 0.05±0.04 | **0.41±0.02** |
| RPSK | 0.67±0.06 | 0.49±0.06 | **0.80±0.03** |
| 3 vs 1 w/ keeper | 0.23±0.01 | 0.20±0.09 | **0.81±0.02** |
| Corner | -0.10±0.33 | -0.06±0.08 | **0.03±0.10** |
| Counterattack easy | 0.00±0.00 | 0.01±0.01 | **0.39±0.02** |
| Counterattack hard | 0.00±0.00 | 0.01±0.02 | **0.53±0.09** |
| 11 vs 11 w/ lazy Opp. | 0.46±0.21 | 0.33±0.07 | **0.72±0.09** |

Table A9: GFootball experiments in *final time metrics:* Average evaluation scores achieved given limited training time. The time limit for each experiment is set to the time when IMPALA finishes training for 5M steps RSK: run to score w/ keeper, PSK: pass, shoot, w/ keeper, RPSK: run, pass, shoot, w/ keeper.

| Method | IMPALA | PPO | Ours (PPO) |
|---|---|---|---|
| Empty goal close | 1.7/2.6 | 5.4/15.5 | **1.0/2.0** |
| Empty goal | 8.4/11.7 | 12.8/19.2 | **2.0/3.9** |
| Run to score | 27.0/34.6 | 16.2/32.5 | **6.3/11.4** |
| RSK | 52.3/- | 51.2/68.2 | **11.5/18.8** |
| PSK | -/- | 70.0/- | **38.8/-** |
| RPSK | 22.3/**25.4** | 45.2/90.8 | **13.5**/27.1 |
| 3 vs 1 w/ keeper | -/- | 67.4/144.2 | **15.9/25.6** |
| Corner | -/- | -/- | -/- |
| Counterattack easy | -/- | 223.2/- | **91.3/-** |
| Counterattack hard | -/- | 383.4/- | **61.8/-** |
| 11 vs 11 w/ lazy Opp. | 58.2/- | 95.8/260.9 | **14.4/72.1** |

Table A10: GFootball experiments in *required time metrics:* required time (minutes) to achieve goal scores (time required to achieve score 0.4 / time required to achieve score 0.8). '-' indicates that the method did not achieve the desired score after 5M environment step training. RSK: run to score w/ keeper, PSK: pass, shoot, w/ keeper, RPSK: run, pass, shoot, w/ keeper.

Figure A3: Atari: **Time** versus reward.

Figure A4: Atari: **Environment step** versus reward.

Figure A5: GFootball: **Time** versus reward.

Figure A6: GFootball: **Environment step** versus reward.