[Reviews · NeurIPS 2020]

Review 1

Summary and Contributions: The high throughput and stable training are exclusive for most parallel actor-learner methods in reinforcement learning, however, this paper proposes a synchronous training scheme to make a balance between the two factors. The method can learn and rollouts concurrently, meanwhile it claims that it can avoid “stale policy”, which often leads to unstable training. The approach is evaluated on Ataris games and Google Football Environment, the results show that this scheme has competitive throughput and higher rewards.

Strengths: The topic is relevant and significant for the RL community, distributed RL or scaling RL is important research directions, which can make RL available for more complicated environments and also shorten the training time. I like the engineering techniques and ideas in the proposed training schema, compared to most previous methods, this method furtherly decouples the function of env step and actor forward. It considers the step time variance of all env steps, which is indeed to be an issue in practical parallel training schemes. Besides, it makes sure the latency between target and behavior policy is fixed to one updating period and adopts adelayed gradient to remedy the latency. The experiment is sufficient with various environments and ablation study, the configuration and details are presented, the implementation code is also uploaded, I am confident with the reliability and reproducibility of the experimental results.

Weaknesses: Although the experiment is fairly solid, I still want to see more experimental comparisons with the typical scaling training method, such as Seed RL[1], since the Seed Rl is also evaluated on the Google Football Environment.

Correctness: In the section of 4.2, the author supposes that X_i^(j) conforms to the distribution of Exp(\beta), and the data consumption of learners also follows the exponential distribution, so are these assumptions suitable and truthful?

Clarity: Yes, this paper is well-written and well-organized, I think I can clearly get the points of the author.

Relation to Prior Work: Some of the ideas (coupling the step function and forward function) are similar to the Seed RL work of Google, but I didn’t see the author mention that and give no experimental comparison.

Reproducibility: Yes

Additional Feedback: * The caption of Figure 3 and 4 is confusion, the text composition is disorder. Updated after author rebuttals: Thanks for adding the comparison to SeedRL, I keep my score for this paper.


Review 2

Summary and Contributions: The paper proposes a new engineering approach to combine the benefits of asynchronous and synchronous RL, while avoiding some of their pitfalls. They describe a method, HTS-RL, which they claim allows for multiple environment interactions and learning updates to happen in parallel, while ensuring deterministic training behavior. They also describe the problem of “stale policies” as a possible explanation for training instability, and propose a fix for it. The architecture has several key features, such as: - batch synchronization: actors are synchronized every \alpha>1 steps, this helps in environments with high variance in terms of the environment step. - concurrent rollout and learning - Using two separate data storages to ensure that the behavior policies are only one step ahead of the target policies, thus avoiding the off-policy issue in some of the existing asynchronous RL approaches.

Strengths: + This is a good engineering step towards better utilizing available compute resources to run deep RL. The approach avoids the low throughput of synchronous RL, while hedging against stale policy updates that plagues asynchronous RL. + They run experiments on several Atari and Google Research Football domains, and show that across most domains, their method: Achieves a higher Atari score in a fixed amount of time Takes less time to achieve a fixed score in GFootball

Weaknesses: - While this is an impressive effort, I am concerned about the limited scientific contribution of this work. Clearly, this is a fairly useful results for practitioners who want to best utilize their available compute power, but there is very limited scientific contribution.

Correctness: Yes.

Clarity: One criticism about how some of the figures are structured: Figure 1 comes very early in the paper. Problem is, there are several undefined entities in that figure (such as executors, polls, etc) that only become known later in the paper. But then, when the figure is referred to in the context describing HTS-RL, I had to scroll back and forth to make sense of the architecture.

Relation to Prior Work: I give the paper a lot of credit in terms of situation themselves relative to existing work, such as A2C, A3C, IMPALA, etc.

Reproducibility: Yes

Additional Feedback:


Review 3

Summary and Contributions: The authors propose a new synchronous deep RL framework to make the best use of hardware while avoiding the stale policy issue via delayed gradient updates.

Strengths: As far as I know, the delayed gradient update and the decoupling of actors and executors are novel in deep RL frameworks. Those concepts can possibly be used in a wide range of RL implementations and eventually make deep RL more accessible. The evaluation metric is convincing and the results support the claims.

Weaknesses: 1. The baselines are somehow weak. Though TorchBeast is a strong baseline, the PPO and A2C from Kostrikov seem weak. As far as I know, faster training is not the goal of Kostrikov's implementation. For PPO, the implementation from OpenAI baselines are stronger, which features parallelization with MPI and all-reduce gradients. For A2C, one could consider rlpyt (rlpyt: A Research Code Base for Deep Reinforcement Learning in PyTorch), where various sampling schemes (including batch synchronization) and optimization schemes can be used. The paper could benefit a lot from a comparison with OpenAI baselines and rlpyt. 2. The explanation of the framework is not easy to follow. In particular, understanding how the two storages and the delayed gradient updates interact with each other is not straightforward. The paper could benefit a lot from a pseudo code, as well as a more detailed Figure 2d additionally showing the flow of parameters and gradients. 3. The utility of batch synchronization As far as I understand, batch synchronization could help if the following assumption holds: Let X_k be the time that an executor needs to run k steps, then Var(X_k) decreases as k increases. There are indeed exceptions, e.g., we could expect that one executor needs 10s to run each of the next 4 steps, while the other executors needs only 1s. In this case, I think batch synchronization won't help. Assuming X_i^(j)s are i.i.d. in Claim 1 seems impractical and the paper could benefit from explicitly stating the above assumption (if I understand it correctly) and clarify this more 4. # of envs v.s. # of threads (cpus) From Line 281, I assume in Figure 4(right), # of envs = # of threads. But this is not a fair comparison for PPO. In openai baseline ppo implementation for Atari games, one thread could have up to 128 envs. In that implementation, it's expected that SPS doesn't change much w.r.t. the # of envs. But I do expect SPS scales linearly w.r.t. the # of threads given the use of MPI and all-reduce gradients. The paper could benefit a lot from studying the performance v.s. # of threads for all the compared algorithms, which would be very useful for practitioners. 5. Credit assignment in Sec 4.1 If I remember it correctly, the two storage feature is similar to the double replay buffer in rlpyt, batch synchronization is also used in rlpyt. I do believe delayed gradient and asynchronous actors and executors are novel. So the paper could benefit a lot by explicitly clarifying what is novel and what is not for the features listed in Sec 4.1 6. The word "updates" in Line 197 is confusing If I understand it correctly, each learner performs only one update -- if there are multiple updates, the delayed gradient updates are not delayed by only 1 step. As you have already pointed out, an update is a forward and a backward pass, I feel it might be better to avoid the use of "updates". I feel it just accumulates gradients but updates parameters only once.

Correctness: Yes

Clarity: Yes

Relation to Prior Work: Yes

Reproducibility: Yes

Additional Feedback: I read the author response and would like to keep my score. I think the authors should use the MPI version of PPO (i.e., 'ppo1' in openai baselines) to make Figure 4 (right) a fair comparison. I do expect the authors' method can still outperform the MPI version of PPO, but the margin should be much smaller.


Review 4

Summary and Contributions: This paper presents a new implementation methodology for synchronous Deep RL algorithms that rely on on-policy data. The shortcoming they seek to overcome is that asynchronous methods of execution typically manage to run faster and are able to distribute their computation more than synchronous methods, due to the variance in per-step execution times in the environments. However synchronous methods tend to be more reproducible and more data-efficient. This paper presents a middle road by synchronizing the execution every time a batch of data is collected. This setup ensures that the data collection policy and the policy that is updated are 1 update apart, leading to fairly on-policy updates and hence better data efficiency. Reproducibility is ensured by having the pseudo-randomness at the level of the executors. The paper analyses and validates the time it takes by their approach to generate a certain amount of data. They also analyze the latency between the behavior and target policy in asynchronous algorithms. The experiments evaluate this approach against both synchronous and asynchronous algorithms, where they check the performance given a fixed time limit, as well as the time taken to reach a fixed performance level.

Strengths: Increasing the throughput of techniques that are more easily reproducible will be useful to the community at large and this aspiration of the paper is to be commended. The analysis and comparison of synchronous and asynchronous algorithms with respect to the amount of data they are able to generate and the lag or off-policyness they suffer from is a valuable part of this paper. The proof of claims in this respect seem to be correct.

Weaknesses: ** AFTER AUTHOR RESPONSE** The authors addressed my specific concerns regarding the off-policy nature of the updates and determinism in the execution. These points are addressed in the appendix. They seem to be important evidence supporting the proposed modification, however, and I encourage them to highlight these findings in the main paper. Given the theoretical and empirical analysis of this practical modification to data gathering, along with discussions with reviewers, I lean more positively regarding this paper. I have updated my score accordingly. _____________________________________________________________________ 1. While the practical benefit to the community due to the above points is clear, I am not completely sure about the scientific value the paper brings. The insights due to the analysis seem fairly minor. Algorithmic innovation seems fairly minor as well. 2. The paper uses samples from a policy that is slightly off-policy without any mention of correcting for this off-policyness, or acknowledging the off-policy update. 3. One of the reasons for introducing synchrony was to ensure determinism in execution. An experiment to showcase this determinism would have been useful.

Correctness: Nothing in the claim proofs or empirical evaluation methodology raises red flags for me.

Clarity: The paper is well written. The authors present the reason for the bottleneck in current synchronous algorithms quite clearly. Figure 1 (e) is not that easy to understand before reading section 4.1, though it is not a major problem.

Relation to Prior Work: Prior work has been presented clearly. The relation to prior work is discussed at various points in the paper. It is both sufficient and clear.

Reproducibility: Yes

Additional Feedback:

[Author Response · NeurIPS 2020]



Figure 1: 11 vs. 11 easy.

Figure 2: Empirical sync. time.

| Method | Kostrikov | OpenAI Baselines | rlpyt | Ours |
|--------|-----------|------------------|-------|------|
| BankHeist | 1382 $\pm$6 | 991 $\pm$14 | 1737 $\pm$39 | **2111 $\pm$21** |
| Beam Rider | 1663 $\pm$14 | 1081 $\pm$18 | 2086 $\pm$32 | **2586 $\pm$14** |
| Breakout | 1225 $\pm$12 | 829 $\pm$31 | 1508 $\pm$60 | **1885 $\pm$15** |
| Frostbite | 1337 $\pm$8 | 962 $\pm$15 | 1803 $\pm$17 | **1973 $\pm$24** |
| Jamesbond | 1353 $\pm$5 | 1014 $\pm$1 | 1991 $\pm$24 | **2139 $\pm$31** |
| Krull | 1443 $\pm$6 | 1057 $\pm$11 | 2001 $\pm$29 | **2657 $\pm$16** |
| KFMaster | 1532 $\pm$15 | 1056 $\pm$8 | 1979 $\pm$55 | **2483 $\pm$15** |
| MsPacman | 1574 $\pm$9 | 1052 $\pm$3 | 1972 $\pm$13 | **2364 $\pm$5** |
| Qbert | 1232 $\pm$13 | 953 $\pm$7 | 1621 $\pm$43 | **1860 $\pm$6** |
| Seaquest | 1593 $\pm$10 | 946 $\pm$21 | 1918 $\pm$25 | **2633 $\pm$32** |
| S. Invader | 1514 $\pm$20 | 1010 $\pm$7 | 1899 $\pm$32 | **2318 $\pm$12** |
| Star Gunner | 1622 $\pm$19 | 1110 $\pm$5 | 2066 $\pm$24 | **2616 $\pm$25** |

Table 1: SPS of different implementations of A2C.

**General response:** We thank all reviewers for their comments.

**Reviewer # 1:**

**Q1: Comparison with Seed RL.** We compare HTS-RL with Seed RL (V-trace) on Gfootball '11vs.11 easy'. For a fair comparison, both HTS-RL and Seed RL use 16 parallel environment processes and one GPU. HTS-RL achieves 829 environment steps per second (SPS) while Seed RL achieves 609 SPS. After 20M steps of training, HTS-RL and Seed RL achieve a $3.55 \pm 0.3$ and $1.50 \pm 0.7$ score difference, respectively. The training curve is shown in Fig. 1.

**Q2: Environment step time assumption.** In Claim 1, we assume the step times follow an exponential distribution. Hence the sum of step times (synchronization time) follows a Gamma distribution. We empirically verify this assumption. In Fig. 2, we show the histogram of synchronization time (sum of every 100 step times) on 'academy 3vs1 w/ keeper'. Furthermore, we perform a Kolmogorov-Smirnov goodness-of-fit test, with a significance-level of 0.05 and D-statistics of 0.04. We find the empirical data is consistent with the assumed Gamma distribution.

**Reviewer # 2:**

**Q3: Scientific contribution of this work.** Like all published high-throughput RL frameworks, *e.g.*, A3C, GA3C, IMPALA, and SeedRL, the scientific contribution of HTS-RL is the reduction in training time. This scales RL research to more complex tasks which is important. Also, HTS-RL maintains the advantages of synchronous RL, *i.e.*, data efficiency and reproducibility, while achieving speedups. Additionally, we provide a detailed analysis of the proposed technique. All reviewers agree that practitioners, *i.e.*, researchers, in our community could benefit from HTS-RL.

**Q4: Placement of figures.** We'll rearrange the figures.

**Reviewer # 3:**

**Q5: Compared Baselines & Additional comparisons with OpenAI and rlpyt.** We compare the speed of different versions of A2C, including our HTS-RL, Kostrikov, OpenAI baselines, and rlpyt, on Atari games. For a fair comparison, all methods use 16 parallel environment processes for data collection, and one GPU for model training/forwarding. For rlpyt, we use the most efficient 'parallel-GPU' mode. As shown in Tab. 1, Kostrikov's A2C is a strong baseline, which achieves $1.4\times$ higher SPS than OpenAI baselines. Also, HTS-RL consistently achieves higher SPS than rlpyt.

**Q6: Pseudo code for clarity.** Actual code is part of this submission. We'll add pseudo code.

**Q7: Practicality of *i.i.d.* assumption of environment step time in Claim 1.** We think an *i.i.d.* environment step time assumption is practical for a machine which satisfies the following conditions: 1) CPU cores are homogeneous; and 2) a CPU's speed is not influenced by the workload of other CPUs. Because we run the same environment on homogeneous CPUs (condition (1)), the step times are identically distributed. In addition, each environment doesn't intervene with any other environments, *i.e.*, environments are independent. From condition (2), the speed of each CPU is also independent. Therefore, the step time in each environment is independent.

**Q8: Batch synchronization may not work in some extreme cases.** In the extreme case described by the reviewer (one executor is $10\times$ slower than the other), we agree that any kind of synchronization decreases throughput. However, empirically, we never observed such cases.

**Q9: # of environments vs. # of threads.** In all experiments, each environment runs independently in a single-threaded process. *E.g.*, in Kostrikov's A2C with 16 environments, 16 envs are handled by 16 parallel single-threaded processes. We do not run multiple environments in one thread for any method. We'll clarify use of 'thread' and 'process.'

**Q10: Crediting double replay buffer and batch synchronization.** To our understanding, the sampling schemes in rlpyt differ from our 'batched synchronization.' One may confuse the 'parallel-GPU' sampling scheme of rlpyt with 'batch synchronization.' However, rlpyt's 'parallel-GPU' mode enforces synchronization across workers at **every** environment batch-step. In contrast, batched synchronization synchronizes all environments every $\alpha$ ($\alpha > 1$) steps (L.187-193). rlpyt uses the double replay buffer in asynchronous mode to increases throughput [A]. In contrast, as discussed in L.181-186, we use a double replay buffer to avoid a stale policy and to maintain determinism in synchronous training. We'll clarify.

**References:** [A] Stooke et al. rlpyt: A Research Code Base for Deep Reinforcement Learning in PyTorch, arxiv. 2019

**Reviewer # 4:**

**Q11: Scientific contribution of this work.** Please see Q3.

**Q12: Addressing the off-policy updates?** We address the off-policy update in Sec. 4.1 (L.202-215) via a one-step-delayed gradient. In addition, in Appendix C, we show that under assumptions, one-step-delayed gradient has the same asymptotic convergence rate as the zero-delayed case.

**Q13: Experiment demonstrating determinism.** Please see Tab. A1 and L.474-475 in the appendix. Even with a different number of actors, the final average scores obtained by HTS-RL are exactly identical. This demonstrates the deterministic nature of HTS-RL.

[Meta-Review · NeurIPS 2020]

This paper proposes a synchronous training scheme for reinforcement learning which address issues with existing synchronous methods (low throughput) and existing asynchronous methods (unstable, non-reproducible, etc.). The reviewers viewed this more of an engineering paper, but the design, execution, and experiments are solid, so we are recommending acceptance. I saw that the paper mentions that code will be released, but I want to emphasize the importance of this, as a large part of the value here is in enabling others to build on and use the proposed method.